# Ibogaine-Mediated ROS/Antioxidant Elevation in Isolated Rat Uterus Is β-Adrenergic Receptors and K_ATP_ Channels Mediated

**DOI:** 10.3390/antiox10111792

**Published:** 2021-11-09

**Authors:** Nikola Tatalović, Teodora Vidonja Uzelac, Zorana Oreščanin Dušić, Aleksandra Nikolić-Kokić, Mara Bresjanac, Duško Blagojević

**Affiliations:** 1Department of Physiology, Institute for Biological Research “Siniša Stanković”, National Institute of Republic of Serbia, University of Belgrade, Bulevar Despota Stefana 142, 11060 Belgrade, Serbia; nikola.tatalovic@ibiss.bg.ac.rs (N.T.); teodora.vidonja@ibiss.bg.ac.rs (T.V.U.); zoranaor@ibiss.bg.ac.rs (Z.O.D.); san@ibiss.bg.ac.rs (A.N.-K.); 2LNPR, Institute of Pathophysiology, University of Ljubljana, Zaloška 4, 1000 Ljubljana, Slovenia; maja.bresjanac@mf.uni-lj.si

**Keywords:** ibogaine, K_ATP_ channels, glibenclamide, β-adrenergic receptors, propranolol, antioxidative enzymes, superoxide dismutase, catalase, uterus, contractility

## Abstract

Ibogaine effects are mediated by cellular receptors, ATP depletion followed by ROS production and antioxidant enzyme activity elevation in a dose and time dependent manner. Since the role of K_ATP_ channels and β-adrenoceptors in ROS cellular circuit was established here we explored their role in ibogaine pro-antioxidant effectiveness. Single dose of ibogaine (10 mg/L i.e., 28.8 μmol/L) was applied to isolated rat uterus (spontaneous and Ca^2+^-stimulated) and contractility and antioxidant enzymes activity were monitored during 4 h. Ibogaine increased amplitude and frequency of spontaneous active uteri immediately after addition that was prevented by propranolol (β_1_ and β_2_ adrenoceptors selective antagonists) and glibenclamide (K_ATP_ sensitive channels inhibitor; only frequency) pre-treatment. In Ca^2+^-stimulated uteri, ibogaine decreased both amplitude and frequency after 4 h. Pre-treatment with propranolol abolished ibogaine induced amplitude lowering, while glibenclamide had no effect. In both types of active uterus, ibogaine induced a decrease in SOD1 and an increase in CAT activity after 2 h. In Ca^2+^-stimulated uterus, there was also a decrease of SOD2 activity after 2 h. After 4 h, SOD1 activity returned to the baseline level, but GSH-Px activity increased. Pre-treatment with both propranolol and glibenclamide abolished observed changes of antioxidant enzymes activity suggesting that ibogaine pro-antioxidative effectiveness is β-adrenergic receptors and K_ATP_ channels mediated.

## 1. Introduction

The ibogaine drug (a monoterpenoid indole alkaloid, an organic heteropentacyclic compound and an aromatic ether; PubChem CID: 197060) is extracted from the rain-forest shrub iboga (*Tabernanthe iboga* Baill.), which grows in West Africa. Ibogaine inspires a sense of well-being and has been used as an anti-addiction agent and is involved as a part of alternative medicine [1,2,3]. Its production, transport, sale, and possession are illegal in many countries (including USA, France, Switzerland, for example) or somewhat prohibited or controlled (Canada, Australia, and Israel) but also legal (Brazil, The Netherlands, Gabon, and South Africa). However, it is not listed in the UN International Narcotics Control Board’s Green List, or List of Psychoactive Substances under International Control. Previous reports on ibogaine human health beneficiaries have been ambiguous and opposite [4,5]. Beneficial effects are evident but with serious precaution and medical supervision. However, in our previous experiments on rats treated with per os doses (1 and 20 mg/b.w.) that affected its energetic metabolism and redox balance, no serious toxic and harmful ibogaine effects were shown [6,7]. Here, in parallel, we aimed to explore the mechanisms of its action on ex vivo model that combined ibogaine pharmacological, metabolic, and redox properties.

Ibogaine affects many different neurotransmitter systems simultaneously [4,8,9] and its effects depend on applied dose and tissue distribution of the most sensitive receptor(s) and effector(s). In addition, by mechanism(s) that are not yet known, ibogaine depletes cellular ATP reserves, and subsequently stimulates higher energy metabolism and metabolic turnover. Increase in the expression of many proteins was found, among them glycolytic enzymes were identified [10,11]. In yeast, ibogaine treatment increased oxidative metabolism and overall ROS production in a dose dependent manner [10,12]. Increased cellular respiration after ibogaine addition is followed by the production of reactive oxygen species (ROS), but significant drop in the total oxidative load on the cell and the elevation of antioxidant enzymes activity [11,12,13]. Since ibogaine is not antioxidant per se, its physiological effect was defined as pro-antioxidative [12,13]. How these ibogaine effects are achieved is unknown, but previous studies in yeast and erythrocytes showed that these effects of ibogaine are not mediated by receptor binding but intrinsic cellular ROS pathways [11,13].

Our previous results on isolated uterus ex vivo have shown that treatment with increasing dose of ibogaine abolished contractile activity and elevated antioxidant enzymes activity by posttranslational mechanisms [14]. Decrease in the contractile activity of isolated uteri could be due to ibogaine-related ATP depletion, alternated ATP turnover, and ROS elevation, especially hydrogen peroxide. Low ATP levels and the depletion of energetics could lead to impairment of rhythms in contractile tissues and reversible contractions ceasing [15]. These effects are under partial purinergic receptor signaling and control that involves also H_2_O_2_ [15,16]. Furthermore, production of ROS (that follows ATP depletion) and its intracellular circulation involves different steps that include K_ATP_ channels [17]. Taken all together, ATP depletion, purinergic receptor, and K_ATP_ channels signaling pathways influence Ca^2+^ flux [18], which can mediate ibogaine induced changes in the contractile activity of uterine smooth muscle, change cellular redox homeostasis, and provoke antioxidant response. Purinergic adenosine A1ARs receptors heterodimerize with β1- and/or β2-adrenergic receptors (β1R and β2R) [19] making functional couple. β2-adrenoceptor coupled with P2Y receptors stimulates release of ATP, leading to Ca^2+^ mobilization from intracellular stores [20]. To all that, combined administration of β-adrenergic and P2X receptors agonists to the heart increased its contractility [21]. Moreover, studies on a transgenic model with cardiac-restricted β_2_-adrenoceptor overexpression (β_2_-TG) showed β_2_-adrenoceptor/nicotinamide adenine dinucleotide phosphate (NADPH) oxidase/ROS/p38 MAPK signaling pathway that indicated superoxide production as central to the detrimental signaling of β_2_-adrenoceptors [22,23]. Therefore, in this work we studied the role of Ca^2+^_,_ K_ATP_ channels and β-adrenoceptors on ibogaine mediated changes in spontaneous and Ca^2+^-stimulated uterine contractility and antioxidant enzymes activity using either glibenclamide (K_ATP_ sensitive channels inhibitor) or propranolol (β_1-_ and β_2_-adrenoceptors selective antagonists) pre-treatments. Study of the pharmacological and metabotropic effects on isolated rat uterus ex vivo is a suitable experimental model because signal transduction systems, energy status, and redox-mediated mechanisms intersect which is proven from our previous works [24,25,26,27,28,29]. Moreover, our previous results of ibogaine treatment on isolated uterus showed significant effect on redox equilibrium and antioxidant levels that were connected with contractility reduction of uterine muscle [14] that led us to investigate ibogaine mechanisms of action further on this model.

Regarding the fact that ibogaine bioavailability as well as plasma levels after oral administration were higher in female than in male rats [30], we applied a dose of 28.8 μmol/L (i.e., 10 mg/L) that mirrors the tissue concentration after ingestion of traditional high ceremonial doses. This dose is in the relevant range shown by in vivo pharmacokinetics of ibogaine [6,31] as well as in ex vivo experiments [11,13]. The activity of antioxidative enzymes: cytosol copper-zinc containing superoxide dismutase (SOD1), mitochondrial manganese containing superoxide dismutase (SOD2), catalase (CAT), glutathione peroxidase (GSH-Px), and glutathione reductase (GR) were measured in isolated uterine muscles 2 and 4 h after treatments.

## 2. Materials and Methods

### 2.1. Experimental Animals

The study was conducted according to the guidelines of the EEC Directive on the Protection of Animals Used for Experimental and Other Scientific Purposes and was approved by the Ethical Committee for the Use of Laboratory Animals of the Institute for the Biological Research “Siniša Stanković”—National Institute of Republic of Serbia, University of Belgrade, No. 06-10/15. Virgin female Wistar rats (RRID: RGD_13508588), body mass 200–250 g, were housed three per cage at 22 °C, day/night 12 h/12 h, with access to food (rodent laboratory chow made by Veterinarski Zavod Subotica, Serbia) and tap water ad libitum. Experiments were performed on isolated uteri from rats in the estrus phase of estrous cycle which was determined by examination of a daily vaginal lavage [32].

### 2.2. Chemicals

Ibogaine hydrochloride (PubChem CID: 197059) purity 99.9% (Remøgen, Phytostan Enterprises Inc., Montreal, Canada) was provided by Mara Bresjanac of the LNPR project under ARRS Program P3-0171, Slovenia. Glibenclamide (PubChem CID: 3488) and propranolol hydrochloride (PubChem CID: 62882) were purchased from Sigma Aldrich (Taufkirchen, Germany, product numbers G0639 and P0884, respectively).

### 2.3. Isolated Organ Bath Studies

Uterine horns were rapidly excised immediately after decapitation, cleaned of surrounding connective tissue and mounted vertically in an organ bath containing 10 mL of De Jalon’s solution at stabile temperature of 37 °C, aerated with carbogen (95% oxygen and 5% carbon dioxide mixture) under the tension of 1 g. Composition of De Jalon’s solution was: NaCl 9.0, KCl 0.42, Na–HCO_3_ 0.5, CaCl_2_ 0.06, glucose 0.5 (in g/L). We used two types of uterine contractile activity in our experiment: spontaneous or stimulated by 0.66 g/L CaCl_2_ to distinguish possible involvement of Ca^2+^ in mechanisms of ibogaine mediated action. After an equilibration period, during which uteri have achieved stable contractile activity, single dose of ibogaine (28.8 μmol/L) was added and contractile activity was monitored up to 4 h. Contractile activity before any treatment served as control (100% of contractile activity). In order to explore the role of K_ATP_ sensitive channels and β_1_- and β_2_-adrenoceptors on the ibogaine mediated effects, another two groups were pre-treated with single dose of either K_ATP_ sensitive channels inhibitor, glibenclamide (Glib) or, the β_1_- and β_2_-adrenoceptors selective antagonists propranolol (Prop) in concentrations of 1 mg/L and 5 mg/L respectively, which per se had no effect on contractions of isolated uterus. Tension of myometrium was recorded in isometric conditions using TSZ-04-E four channel isolated organ bath system equipped with FSG-01 force displacement transducers and EXP-CLSG-4 four channel bridge amplifier (MDE Research, Heidelberg, Germany). Data recording and analysis were performed using SPEL Advanced Kimograph v2.94 software (Experimetria Ltd. and LogiRex Software Laboratory, Budapest, Hungary).

To study effects on the activity of antioxidant enzymes, treated isolated uteri were taken from each experimental group, as well as untreated control samples, after 2 and 4 h of contractile activity and frozen immediately in liquid nitrogen and transferred to −80 °C until the measurement of antioxidant enzyme activities.

### 2.4. Measurement of Antioxidant Enzyme Activities

Frozen uterine horns were thawed, homogenized (3 × 10 s) and sonicated (3 × 15 s, at 10 kHz) in buffer containing: 0.25 M sucrose, 0.05 M Tris and 1 mM EDTA, pH 7.4. Sonicates were than centrifuged (Optima^TM^ L-100 XP Ultracentrifuge, Beckman Coulter, California, CA, USA) 90 min at 105.000× *g* at 4 °C, and the supernatants were used for spectrophotometric measurement of enzyme activities using a Shimadzu UV-160 spectrophotometer (Shimadzu Scientific Instruments, Japan). The activities of total SOD and SOD2 were determined by the adrenaline method monitoring the absorbance at 480 nm [33]. A SOD activity unit (U) is defined as the amount of enzyme needed for 50% decrease in the adrenaline autoxidation rate at pH 10.2. SOD2 activity was measured after inhibition of SOD1 by incubating the samples with an equal volume of 8 mM KCN. SOD1 activity was calculated by subtracting SOD2 from total SOD activity. CAT activity was measured by monitoring H_2_O_2_ consumption at 230 nm [34]. Glutathione peroxidase (GSH-Px) activity was measured according to modified assay described by Paglia and Valentine [35] by monitoring NADPH consumption at 340 nm. Tert-Butyl hydroperoxide was used as a substrate. One unit of GSH-Px activity was defined as the amount required to oxidize 1 nM NADPH per min at 25 °C and pH 7.0. The activity of GR was measured using the method based on NADPH oxidation and GSSG reduction, monitoring the decrease in absorbance at 340 nm [36]. One unit of GR activity is defined as the oxidation of 1 nM NADPH per min at 25 °C and pH 7.4. Enzyme activities were expressed as U per milligram of protein.

### 2.5. Data Analysis and Statistical Procedures

Statistical analyses of the results were performed following the protocols described by Hinkle et al. [37]. The following parameters of contractile activity were analyzed: amplitude (calculated as percentage of controls) and frequency (expressed as a number of contractions per hour). Each data value is expressed as the mean ± SEM. Statistical hypothesis testing was performed using parametric tests since the data were in accordance with required assumptions. Normality of data samples distribution was tested by Shapiro–Wilk test. The effect of ibogaine on amplitude and frequency of uterine contractility was tested by two-way ANOVA and post hoc compared by Tukey’s HSD. Frequency values i.e., percent were previously logarithmically (ln) transformed. The effect of single dose of ibogaine on amplitude and frequency of uterine contractility was tested for the following factors: type of contractions and time, while the effects of pre-treatment with propranolol and glibenclamide were tested for factors: treatment and time. The activities of antioxidant enzymes were compared by two-way ANOVA (factors: treatment and time) followed by Tukey’s *HSD* post hoc test (significance: *p* < 0.05).

## 3. Results

### 3.1. Effects of Single Dose of Ibogaine on Uterine Contractions

Single dose of ibogaine (28.8 μmol/L) increased amplitude of spontaneous contractions immediately after addition. Amplitude levels were elevated in spontaneously active uterine muscle one hour after ibogaine addition staying at that level for 2 h (*p* < 0.01). During the 4th hour, the amplitude of spontaneously active uteri treated by ibogaine was at the level similar to those recorded at the beginning of the experiment (ANOVA, time effect *p* < 0.001, *post hoc* test *p* < 0.01; Figure 1a,b). However, addition of a single dose of ibogaine (28.8 μmol/L) to Ca^2+^-stimulated uteri had no effect immediately after addition but led to a decrease in amplitude magnitude after 4 h (ANOVA, time effect *p* < 0.001, TxH interaction effects *p* < 0.01; post hoc test *p* < 0.05; Figure 1a,b).

Ibogaine accelerated frequency of spontaneously active uteri immediately after addition (Tukey’s HSD post hoc comparison; controls vs. ibogaine first and second hour, *p* < 0.01) and high frequency persisted during the experiment (ANOVA time effect non-significant, Figure 1a,c). However, addition of ibogaine decreased the frequency of Ca^2+^-stimulated active uteri and finally almost ceased at 4th hour (two-way ANOVA, TxH interaction effect, *p* < 0.001, post hoc *p* < 0.01; Figure 1a,c).

### 3.2. Effects of Single Dose of Ibogaine on Uterine Contractions: The Effect of Glibenclamide and Propranolol Pretreatment

Pre-treatment with propranolol prevented the increase of amplitude of contractions after ibogaine treatment of spontaneously active uteri (ANOVA treatment effect, *p* < 0.001, interaction TxH, *p* < 0.001; Figure 2a). Furthermore, amplitude level was significantly lower at the 4th hour of activity of propranolol pre-treated uteri compared to ibogaine (ANOVA Time effect, *p* < 0.001, post hoc Tukey’s HSD, *p* < 0.001; Figure 2a). There was no effect of glibenclamide pre-treatment on spontaneous uterine contractile activity after a single ibogaine dose.

Pre-treatment with both propranolol and glibenclamide prevented the increase of contraction frequency induced by ibogaine single dose (ANOVA treatment effect; *p* < 0.001; Figure 2b). At the end of experiment, pre-treatment of uteri additionally decreased frequency of ibogaine stimulated spontaneous active uteri (ANOVA time effect, *p* < 0.001) being more pronounced in glibenclamide pre-treated uteri (ANOVA time and TxH interaction effect, *p* < 0.001).

Propranolol prevented ibogaine induced lowering of the amplitude level 4 h after ibogaine single dose addition in Ca^2+^-stimulated active uteri (ANOVA treatment effect, *p* < 0.01; time effect, *p* < 0.001; TxH interaction effect, *p* < 0.05: post hoc Tukey’s HSD, *p* < 0.01; Figure 3a). There was no effect of glibenclamide pre-treatment on amplitude levels of Ca^2+^-stimulated uterus after ibogaine single dose addition.

On the other hand, pre-treatment with propranolol and glibenclamide had no statistically significant effect on frequency of Ca^2+^-stimulated uterine contractility treated with single dose of ibogaine. Regardless of pre-treatment, ibogaine treatment led to a decrease in frequency after 4 h (ANOVA time effect, *p* < 0.001; Tukey’s HSD, 4th hour vs. beginning of the experiment, *p* < 0.05).

### 3.3. Effects of a Single Dose of Ibogaine on Antioxidant Enzymes Activity: The Effect of Propranolol and Glibenclamide Pre-Treatment

Treatment of spontaneously active uteri with a single dose of ibogaine decreased SOD1 activity (*p* < 0.001) and increased CAT activity (*p* < 0.01) after 2 h (Figure 4 and Figure 5). After 4 h, the level of SOD1 was as in controls, CAT activity was still increased (*p* < 0.001), and GSH-Px increased (*p* < 0.05). The difference in the activity of GR between controls and ibogaine treated was not statistically significant.

On the other hand, in Ca^2+^-stimulated active uteri, after 2 h both SOD1 and SOD2 activities decreased (*p* < 0.001; Figure 4 and Figure 5). After 4 h, SOD2 activity was still decreased compared to the controls (*p* < 0.01), while SOD1 was at the level of controls. CAT activity increased after 2 h of ibogaine addition (*p* < 0.01). After 4 h, CAT (*p* < 0.001) and GSH-Px (*p* < 0.001) were elevated. As in spontaneously active uteri, the activity of GR appeared to be decreased, but the difference was again not statistically significant.

Results of the measurement of antioxidant enzymes activity showed that propranolol pre-treatment abolished all ibogaine induced changes of antioxidant enzymes activity (Figure 4), except SOD2 after 2 h (ANOVA significant TxH interaction, *p* < 0.001; post hoc Tukey’s HSD, *p* < 0.001). However, addition of propranolol alone elevated the activity of SOD2 comparing to controls (*p* < 0.01 measured after 2 h, and *p* < 0.05 after 4 h). In Ca^2+^-stimulated uteri, addition of propranolol decreased SOD1 (*p* < 0.001) and elevated CAT (*p* < 0.01) activity after 2 h, but levels were as in controls after 4 h.

Glibenclamide also abolished all ibogaine induced changes of antioxidant enzyme activities in both types of active uteri (Figure 5). Glibenclamide addition also had effect on antioxidant enzymes activity in active uteri. In spontaneously active uteri, only SOD2 activity increased after 2 h (*p* < 0.05) after propranolol addition, but in Ca^2+^-stimulated SOD1 decreased after 2 h (*p* < 0.01) and SOD2 after 4 h (*p* < 0.05).

## 4. Discussion

A large number of experiments (in vitro, in vivo, and ex vivo) and human studies indicate a complex pharmacology of ibogaine that affects many different neurotransmitter systems simultaneously [4]. Its effects depend on applied dose and tissue distribution, with the most prominent effects involving the most sensitive receptor and effector(s). Sigmoid response of uterine contractility to an increasing dose of ibogaine in a hormetic manner found in our previous work [14] was in concordance with observed effects at traditional use of iboga, where small doses were used for its stimulant properties while higher doses used in ritual context were praised for their inhibitory influence on brain cortex causing shifts in perception with emergence of subconscious contents. However, increasing the addition of ibogaine led to cessation of uterine contractility ex vivo which could possibly be attributed to different receptors and eventually a reversible ATP and/or Ca^2+^ depletion. In a single dose, ibogaine stimulated utilization and releasing of ATP right after addition and increases its re-synthesis, oxidative metabolism and respiration followed by the increase of cellular reactive oxygen species [12,13]. Treatment with single oral dose of ibogaine in vivo showed higher glycogenolytic and mitochondrial activity accompanied by increased hepatic xanthine oxidase activity suggesting faster adenosine turnover [6,7]. Therefore, increased contractile activity observed in spontaneously active uteri found in our experiment can be a manifestation of the redirection of ATP consumption after ibogaine addition, since ATP is an internal excitatory transmitter [38] that induces muscular contraction coupled to differential signal pathways leading to intracellular Ca^2+^ increase and mobilization [15]. Contractions are normally stable due to interactions between the sarco-endoplasmic reticulum (SER) and mitochondria [39,40,41,42] and Ca^2+^ shuttling between the two organelles [43]. Previous results showed that ceasing of contractions was not due to the ATP depletion, as the effects occurred independently of an ATP synthase inhibitor [44]. ATP mediated contractions persisted until the depletion of intracellular Ca^2+^ stores as well as in the absence of extracellular Ca^2+^ [38]. Contractions of spontaneous active uteri after 4 h upon ibogaine addition are still higher than non-treated control and ibogaine effects can be considered as proactive and stimulatory. Ibogaine effects related to P2X-purinoceptors were noted as well as a concentration-dependent enhancement of spontaneous contractions [45]. P2X receptors mediate ATP dependent activation leading to Ca^2+^-sensitive intracellular processes suggesting another mechanism which can be operative after ibogaine addition that stimulates spontaneous uterine activity. Stimulation can also come from ATP β_2_-adrenoceptor coupled with P2Y receptors stimulated release, leading to Ca^2+^ mobilization from intracellular store. Pre-treatment with propranolol abolished the stimulatory effect of ibogaine in spontaneous active uteri amplitude suggesting that this signaling pathway is operative. Moreover, addition of Ca^2+^ to spontaneously contracting isolated uterine muscle in our experiment seems to diminish the importance of ATP mediated increase of amplitude after ibogaine addition due to the Ca^2+^ stimulatory effect on uterine contractility that overlaps ibogaine stimulatory effect making propranolol ineffective. In spontaneously active uteri, ibogaine addition also elevated frequency which can be multi receptor mediated processes since membrane potential is regulated by different types of receptors, as well as ATP and Ca^2+^. On the other hand, local membrane potential regulates β_2_-adrenergic receptor coupling to Gi3 coordinated with Ca^2+^ [46]. Pre-treatment with both propranolol and glibenclamide prevented ibogaine stimulation of frequency due to blockade of β-adrenergic receptor as well as K_ATP_ channel suggesting that ibogaine frequency stimulation in spontaneous active uteri involves these types of receptors that are under delicate balance of ATP, Ca^2+^, and intrinsic uterine peacemaker activity. Membrane K_ATP_ channels are suggested as couplers of intracellular energetics to electrical activity [47]. In pancreatic cells, when metabolism is high, decreased K_ATP_ channel activity promotes membrane depolarization, and triggers cellular responses [48].

On the contrary, external Ca^2+^ stimulates frequency per se and additional ibogaine that releases ATP led to a decrease in frequency due to some unknown inhibitory mechanism(s). Decrease in contractile activity in Ca^2+^-stimulated active uteri by ibogaine was mainly due to its inhibitory effect on frequency. Effect of pre-treatment was different with respect to external Ca^2+^ concentration. Namely, in Ca^2+^-stimulated uteri pre-treatment with either propranolol or glibenclamide did not prevented ibogaine induced frequency attenuation. When uteri were stimulated with external Ca^2+^, ibogaine mediated decrease of contraction force was attenuated by maintenance of amplitude by propranolol, suggesting a blockade of adrenoceptors as protective in excessive ibogaine and Ca^2+^ stimulation of uterus. Since the induction of uterine activity by addition of Ca^2+^ establishes levels that enable stable contractions and intercellular Ca^2+^, ATP redirection seems ineffective after ibogaine addition in Ca^2+^-stimulated uteri. Due to the pharmacological pluripotent nature of ibogaine molecule and its metabolic conversion to an active metabolite noribogaine [49] direct involvement of other receptors in regulation of uterine contractility are possible. According to previous results [11], within the time needed for depleted ATP levels (about 1 h) attenuation of contractility in Ca^2+^-stimulated active uteri occurs. It is possible that other ibogaine mediated changes of cellular effectors led to the decrease of contractility. One candidate is H_2_O_2_ since it was shown to mediate attenuation of contractility [26]. H_2_O_2_ mediated contractility fading was concentration dependent and involved voltage-gated potassium channels [27]. Our results here showed that in active uteri 4 h after addition of ibogaine, both GSH-Px and CAT activities increased, suggesting a very high impact of cellular H_2_O_2_. According to the levels of antioxidant enzymes, H_2_O_2_ is elevated in both spontaneous and Ca^2+^-stimulated active uteri after both 2 and 4 h of treatment when a decrease of uterine activity occurs. However, this decrease is significant only in Ca^2+^-stimulated uterus, suggesting greater and detrimental impact of H_2_O_2_, but still preserved contractility. In our previous work [14] we have shown that intensity and force of contractions have been several times higher in Ca^2+^-stimulated uterus. The same holds for levels of antioxidant enzymes (SOD1, SOD2, and CAT) suggesting that both production and elimination of H_2_O_2_ were higher compared to spontaneously active uteri. Although higher, levels of antioxidant enzymes in Ca^2+^-stimulated uteri seem insufficient to prevent fading of contractions after ibogaine addition and harmful effects of H_2_O_2_. Results also showed that SOD1 activity was decreased after 2 h in both types of uterine activity again suggesting a role of H_2_O_2_ since it is known that SOD can be inhibited by H_2_O_2_ [50]. In Ca^2+^-stimulated uteri, SOD2 is also inhibited suggesting prevention of excessive production of mitochondrial H_2_O_2_. Based on uterine antioxidant levels, intensity of oxidative processes after a single dose of ibogaine was very high, being higher in Ca^2+^-stimulated uteri, but with different dynamics. These results are in accordance with our previous results that showed ibogaine dose dependent elevation of antioxidant enzymes activity [14] that was not a consequence of elevated protein synthesis, but posttranslational modifications.

Bearing in mind that ibogaine redox activity is not executed by its direct binding to receptors, we focused on the effect on antioxidant enzymes and cellular receptor mediated ROS/redox processes since ibogaine was suggested to be pro-antioxidative [11,12,13]. We pre-treated uterus with β_1_- and β_2_-adrenoceptors selective antagonists (propranolol) and K_ATP_ sensitive channels inhibitor (glibenclamide) since it was shown that these receptors and ion channels were modulators of cellular ROS circulation. However, since ibogaine express its action on ATP and ROS related metabolic pathways here we showed again that its application significantly elevated antioxidant enzymes activity in a relatively short time. Our results on antioxidant enzymes activity obtained after pre-treatment with propranolol and glibenclamide suggest cellular ROS involved circulation that are mediated by β-adrenergic receptors and K_ATP_ channels. Namely, ATP reacts with purinergic P2Y2 receptors to activate NADPH oxidase homolog dual oxidase 1 (DUOX1)-dependent production of H_2_O_2_ that in turn activates ATP-dependent cell signaling. It was shown that ibogaine works partly through purinergic signaling [45], but that time ibogaine influence on ATP metabolism was unknown. On the other hand, adrenergic receptors are connected with ROS signaling via activation of Nox and subsequent production of superoxide [22,23]. Blockade of adrenergic signaling by propranolol decreases the signaling purpose of H_2_O_2_ production and prevents compensatory elevation of CAT and GPx activities. Moreover, there is a cross-talk between mitochondrial and cytosolic (NADPH oxidase-derived) reactive oxygen and nitrogen species that is regulated by K_ATP_ channel [17,51]. Inhibition of the channel by pre-treatment with glibenclamide resulted in lowering H_2_O_2_ that resulted in decreasing both CAT and GPx activity in ibogaine treated uteri to the level of controls. CAT activity is regulated by the tyrosine-protein kinase c-Abl/Arg pathway that promotes CAT phosphorylation and enhancing activity under elevation of H_2_O_2_ concentration upon certain point [52], and peroxisomal CAT and cytoplasmic GPx-1 coordinate expression and/or downregulation seems to be achieved by FoxO (family of Forkhead transcription factors)-regulated elements [53], suggesting that H_2_O_2_ may be reduced in multiple cellular compartments in parallel. Pre-treatments with either propranolol or glibenclamide blockade of different receptors before ibogaine addition decreased production of H_2_O_2_ and levels of CAT and GPx remained at control levels. Our results suggest that ibogaine pro-antioxidative activity was conducted by intrinsic cellular ROS circulating processes that involve K_ATP_ channels as well as adrenergic/purinergic receptor/Nox mediated signaling. Furthermore, high energetic demands after ibogaine mediated ATP depletion activated intense mitochondrial respiration that included control of membrane potential, ionic exchange and Ca^2+^ regulation processes where mitochondrial K_ATP_ channels plays significant role enabling establishment of ATP gradient in an optimal way. ROS (superoxide and H_2_O_2_) are intrinsic part of that processes. Ibogaine provoked significant ATP and ROS disturbance that was followed by regulation of antioxidant enzymes activity which in turn regulated ROS dynamics toward both optimal H_2_O_2_ cellular signaling and protection against oxidative damage. Changes in antioxidant defense suggest that ibogaine disturbed cellular energetic, redox and ROS homeostasis which is in accordance with previous studies [11,12,14]. In both types of uterine activity studied here (spontaneous and Ca^2+^-stimulated) lower SOD1 (cytosol Cu, Zn containing enzyme) after 2 h and elevated GSH-Px activities after 4 h were registered, suggesting elevation of cellular hydrogen peroxide. It is known that H_2_O_2_ inhibits SOD1 activity suppressing its own production [54,55]. Increased GSH-Px activity could be compensatory to elevated levels of H_2_O_2_. On the other hand, low level of NADPH is a favorable cell environment for the decrease of GR activity [56]. Lower GR activity we found 4 h after single dose of ibogaine could be the consequence of lower NADPH concentration since restitution of ATP occurs. Pre-treatment with propranolol and glibenclamide reversed these changes and, thus, prevented significant alternations in antioxidant enzymes activities suggesting significant involvement of adrenergic receptors and K_ATP_ channels in the regulation of cellular ROS homeostasis and ibogaine action.

Currently there are three ongoing clinical trials that investigate ibogaine treatment of addiction (EU Clinical Trials Register 2014-000354-11; ClinicalTrials.gov Identifier: NCT03380728; ClinicalTrials.gov Identifier: NCT04003948). Our results on isolated uterine ex vivo model showed that β-adrenergic receptors as well as K_ATP_ channels are involved in ibogaine mediated pro-antioxidant action (Figure 6) that can contribute to possible future ibogaine medical use and lead to modification of ibogaine treatment protocols toward greater safety. The use of ibogaine as pro-antioxidant agent in our experiment showed mechanisms of ROS mediated cellular processes and the regulation of antioxidant defense enzymes activity with potential broader significance independent of ibogaine itself.

## 5. Conclusions

Cellular signaling pathways and effector mechanisms of ibogaine mediated redox activity are interconnected with β-adrenergic receptors and K_ATP_ channels pathways which seems to play an important role in cellular ROS circulation and the elevation of antioxidant enzymes activity.

## Figures and Tables

**Figure 1 antioxidants-10-01792-f001:**
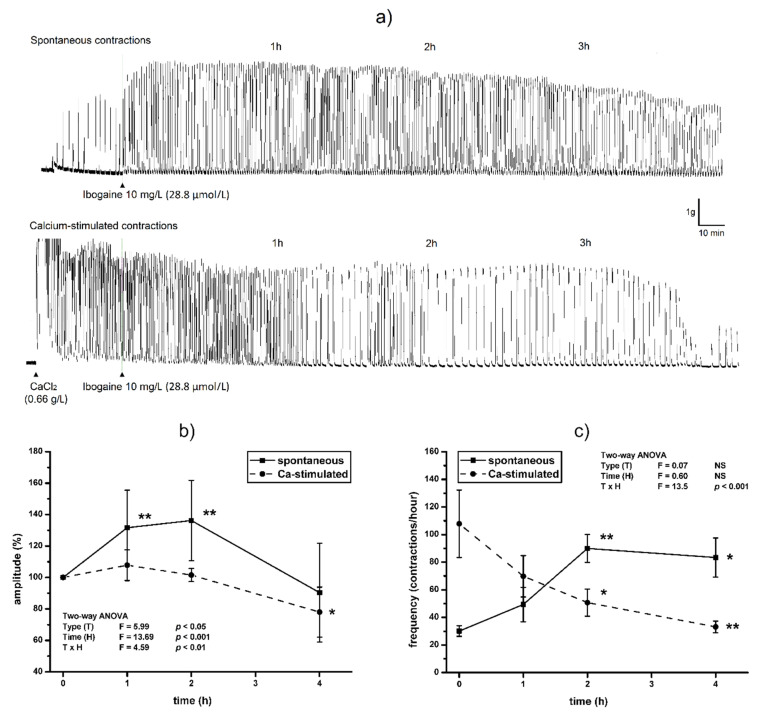
The effect of single dose of ibogaine (28.8 μmol/L) on uterine contractility: (**a**) Representative original traces of spontaneous and Ca^2+^-stimulated active uterus during 4 h, treated with a single dose of ibogaine. (**b**) Uterine amplitude levels during 4 h after a single dose of ibogaine treatment. (**c**) Uterine frequency levels during 4 h after a single dose of ibogaine treatment. Results are expressed as the mean ± SEM (*n* = 7). Statistical significance was tested by two-way ANOVA (factors (F): type of contraction—T and time—H) and post hoc compared by Tukey’s HSD test. * *p* < 0.05; ** *p* < 0.01; NS—non-significant.

**Figure 2 antioxidants-10-01792-f002:**
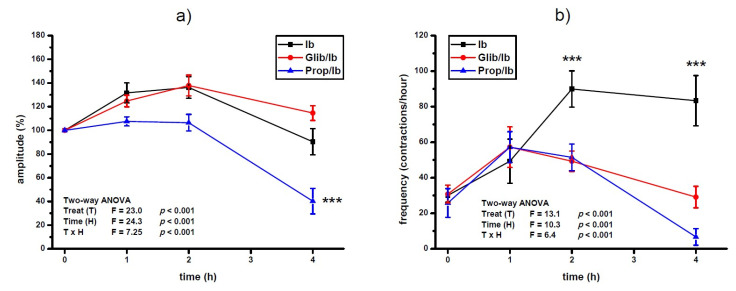
Amplitude (**a**) and frequency (**b**) of spontaneous uterine contractions treated with a single dose of ibogaine (28.8 μmol/L) and pre-treated with either glibenclamide or propranolol. Results are expressed as the mean ± SEM (*n* = 7). Statistical significance was tested by two-way ANOVA (factors: treatment—T and time—H) and post hoc compared by Tukey’s HSD test. *** *p* < 0.001.

**Figure 3 antioxidants-10-01792-f003:**
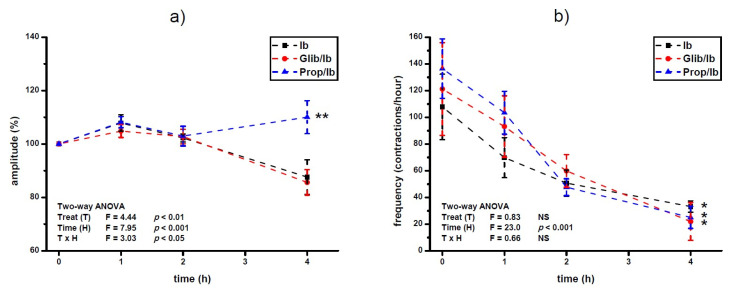
Amplitude (**a**) and frequency (**b**) of Ca^2+^-stimulated uterine contractions treated with a single dose of ibogaine (28.8 μmol/L) and pre-treated with either glibenclamide or propranolol. Results are expressed as the mean ± SEM (*n* = 7). Statistical significance was tested by two-way ANOVA (factors: treatment—T and time—H) and post hoc compared by Tukey’s HSD test. * *p* < 0.05; ** *p* < 0.01; NS—non-significant.

**Figure 4 antioxidants-10-01792-f004:**
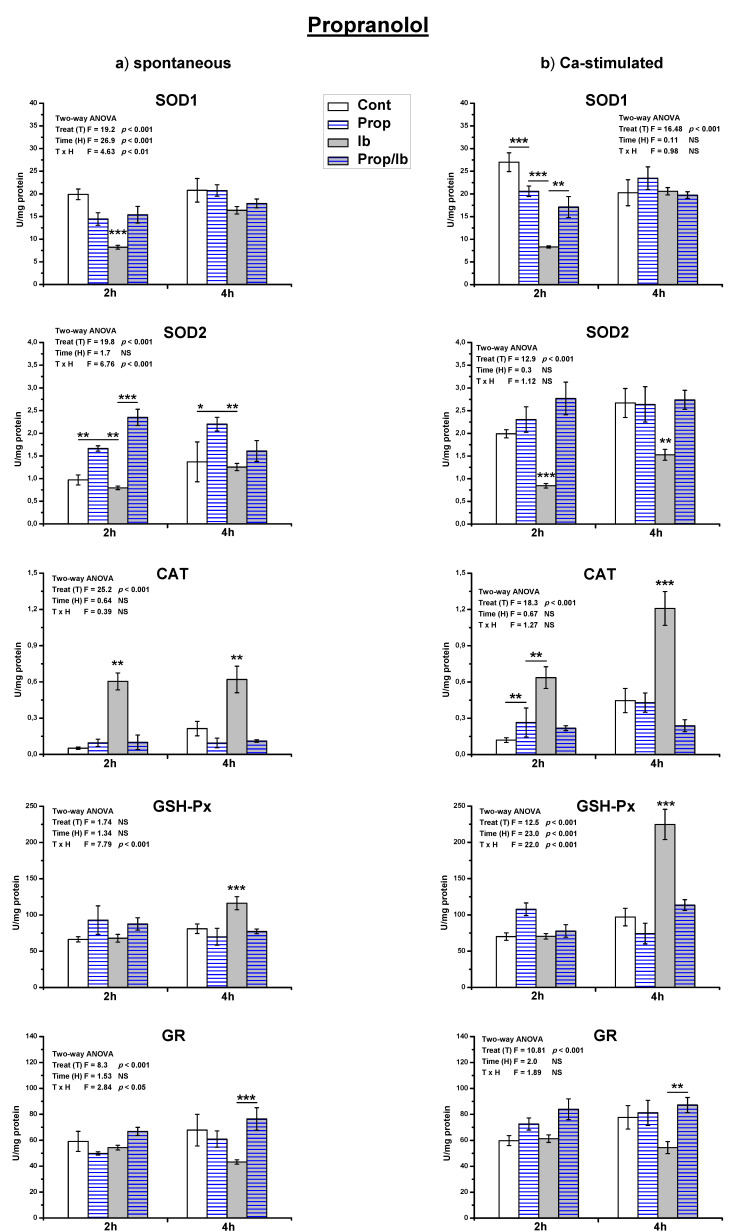
The activity of antioxidant enzymes in both spontaneous (**a**) and Ca^2+^-stimulated (**b**) active uteri treated with a single dose of ibogaine (28.8 μmol/L) and pre-treated with propranolol. Data are expressed as the mean ± SEM (*n* = 7). Statistical significance was tested by two-way ANOVA and post hoc compared by Tukey’s HSD test. * *p* < 0.05; ** *p* < 0.01; *** *p* < 0.001.

**Figure 5 antioxidants-10-01792-f005:**
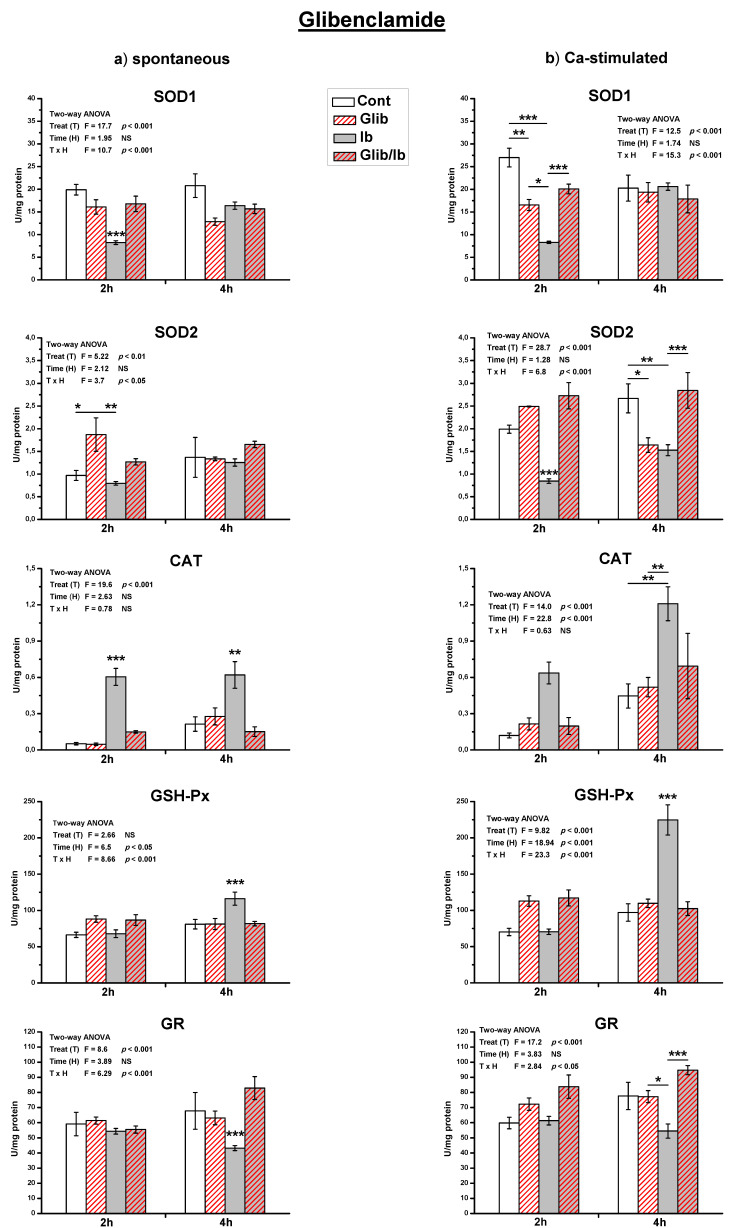
The activity of antioxidant enzymes in both spontaneous (**a**) and Ca^2+^-stimulated (**b**) active uteri treated with a single dose of ibogaine (28.8 μmol/L) and pre-treated with glibenclamide. Data are expressed as the mean ± SEM (*n* = 7). Statistical significance was tested by two-way ANOVA, and post hoc compared by Tukey’s HSD test. * *p* < 0.05; ** *p* < 0.01; *** *p* < 0.001.

**Figure 6 antioxidants-10-01792-f006:**
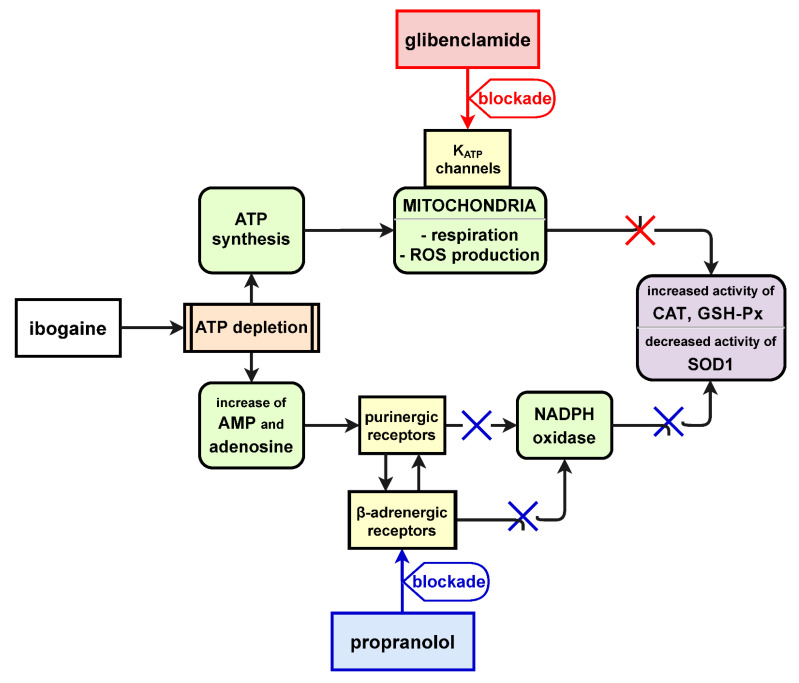
Schematic depiction of proposed role of β-adrenergic receptors and K_ATP_ channels in ibogaine mediated pro-antioxidant action.

## Data Availability

The data used to support the findings of this study are included within the article.

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
