# Peer review of "Ibogaine-Mediated ROS/Antioxidant Elevation in Isolated Rat Uterus Is β-Adrenergic Receptors and KATP Channels Mediated"

_antioxidants, 2021, doi:10.3390/antiox10111792_

Round 1
Reviewer 1 Report
Dear colleagues!The authors' research is of interest for pharmacology and medicine. The paper clearly describes the methods and results, and provides a detailed discussion. The figures are informative, including from the statistical side.However, there are some recommendations and questions.
Author Response
Dear Reviewer,
Thanks for Your very useful comments and suggestions. We used them to improve the manuscript and add some parts.
1) I propose to indicate the legality of ibogaine in different countries.
Ibogaine is not included on the UN International Narcotics Control Board's Green List, or List of Psychoactive Substances under International Control. Since 1989, it has been on the list of doping substances banned by the International Olympic Committee. In many countries possession, sale, transport and production of ibogaine are somewhat prohibited or controlled (Canada, Australia, Israel) or illegal (including USA, France, Switzerland, for example). Exceptions are New Zealand, Uruguay, Slovenia, Netherland… In Serbia, it is listed as psychotropic substance that can cause severe damage to human health. We have no access for many countries, and it is hard job to list them all. We have now indicated legality of ibogaine in the manuscript (in Introduction), but we did not want to burden the text and the results with this matter too much. Ibogaine legality status is a matter of controversy due to heart failure and accidental deaths after its use or treatment. However, there are several publicly announced and advertised treatments that includes ibogaine as health agents in countries where its use is not prohibited.
2) The prospects of using ibogaine for medical purposes?
Previous reports on ibogaine beneficiaries have been ambiguous and opposite. Beneficial effects are evident, but with serious precaution and medical supervision.
Our results did not show any serious toxic and harmful ibogaine effects on rats treated with per os doses that did affected its energetic metabolism and redox balance (Vidonja Uzelac 2019a, 2019b, referenced in the text, and Tatalović et al, 2021 „Ibogaine has sex specific effects on the cellular antioxidant systems in rats: the study on females“ submitted for publication). Here, in parallel, we have been trying to explore the mechanisms of its action on ex vivo model that combined ibogaine pharmacological, metabolic and redox properties that contribute and could lead to its safe use.
There are clinical studies running that will shed more light on this topic.
- The Efficacy of Ibogaine in the Treatment of Addiction; an open label, single fixed dose pilot-study of the efficacy of ibogaine in opioid-dependent subjects
EU Clinical Trials Register 2014-000354-11
https://www.clinicaltrialsregister.eu/ctr-search/search?query=2014-000354-11
- Ibogaine in the Treatment of Alcoholism: a Randomized, Double-blind, Placebo-controlled, Escalating-dose, Phase 2 Trial
ClinicalTrials.gov Identifier: NCT03380728
https://clinicaltrials.gov/ct2/show/NCT03380728
- Preliminary Efficacy and Safety of Ibogaine in the Treatment of Methadone Detoxification
ClinicalTrials.gov Identifier: NCT04003948
https://clinicaltrials.gov/ct2/show/NCT04003948
We included some of above points into Discussion.
3) Please explain the reason for using parametric methods of statistical analysis. It is unclear whether the distribution of signs was normal.
The dependent variables are measured at the continuous level. Each two independent variables consists of two or more categorical, independent groups. There is an independence of observations, which means that there is no relationship between the observations in each group or between the groups themselves. There are no significant outliers. There is a homogeneity of variances for each combination of the groups of the two independent variables.
Since the number of samples (animals) used in the experiment have to be reasonable but statistically enough, we used here from 6 to 9 per experimental point, and took into account that the distribution had been normal in our similar previous experiments (our previously calculations of normality of the distribution of signs on the same experimental model showed that the distribution of signs was normal). We tested for normality using the Shapiro-Wilk test (the sentence added in Material and Methods section). In our experiment specifically, dependent variables are normally distributed for each combination of the groups of the two independent variables, with a few exceptions that ANOVA permit. Namely two-way ANOVA only requires approximately normally distributed data because it is quite "robust" to violations of normality. Thus the assumption of normality can be a little violated and still provide valid results (as is the case with the activity of some antioxidant enzymes activity in our research).
Reviewer 2 Report
Abstract should be more clear and concise
In Introduction and in Discussion elaborate more on
the Ibogaine redox signaling mechanism including all gene interactome with beta receptor and Katp
Author Response
Dear Reviewer,
Thanks for useful comments and suggestions. We try to expand and more elaborate ibogaine mechanisms according to the suggestions.
- Abstract should be more clear and concise
We tried to make abstract more clear and concise. The number of the words for abstract provided by the journal rules is less than usually. We think, now is ok.
- In Introduction and in Discussion elaborate more on the Ibogaine redox signaling mechanism including all gene interactome with β-receptors and Katp channels.
We extended Introduction and Discussion according to your suggestions and try elaborate them more.
Ibogaine redox signaling mechanisms have not been recognized yet. This manuscript is follow-up of previous work that showed ibogaine effects on ATP metabolism, energetic homeostasis, ROS production, antioxidant enzyme activity and, thus, changed redox homeostasis. Some key conclusions from previous results have been extracted such as ATP depletion, elevation of energetic metabolism, and antioxidant activity, but without the exact mechanisms. All of these papers are mentioned in the manuscript and listed in the reference list. Our manuscript seems to be the first one that deals with ibogaine redox signaling mechanisms and shows that are mediated by KATP channels and β-adrenergic receptors that can be prevented by their blockade. Signaling route is further elaborated in Discussion.
Our previous work on ibogaine (Oreščanin Dušić 2018) showed (by Western Blot analysis) that the amount of antioxidant proteins were not changed after 4 hours of experiment and that the elevation of its activity was posttranslational modulated. Gene interactome as well protein-protein interactome are included and discussed in Discussion. In the light of these findings, we specified physical interaction between β-receptors and A1 purinergic receptor (heterodimer formation) and discuss its possible involvement and significance in ibogaine redox signaling mechanisms.
Reviewer 3 Report
The manuscript titled “Ibogaine mediated ROS/antioxidant elevation in uterus is β-adrenergic and KATP receptors mediated” attempted to evaluate antioxidant activities of Ibogaine in rat uterus.
As such the work in this manuscript is interesting. However, there are some deficiencies in the manuscript which need to be addressed to make it publishable:
- The name of the animal model is needed in the title
- What was the rationale of using uterus used as the target tissue?
- The chemical structures of the active ingredients of Ibogaine are needed.
- The objective(s) of the work is not clearly mentioned in the abstract and introduction.
- Figure 4 and Figure 5 are somewhat unclear and clearer pictured will be easier to follow the data by readers.
- A mechanistic figure and related explanations are required to summarize the data.
- What is/are the future work plan with this agent.
Author Response
Dear Reviewer,
Thanks for useful suggestions and comments. We improved the manuscript according to your suggestions.
1 The name of the animal model is needed in the title
The name of the animal model is added in the title.
2 What was the rationale of using uterus used as the target tissue?
The use of uterus was explained with one sentence in the text, but we have extended it. In uterus, very well characterized complex signal transduction systems are present and high abundancy of different types of receptors and ion channels as well as the redox sensitivity of smooth muscle contractility makes it a very suitable experimental model for this kind of study which has been proven from previous works (Appiah 2009, Oreščanin Dušić 2006, Mijušković 2014, 2015). Furthermore, our previous results of ibogaine treatment on isolated rat uterus showed significant effect on redox equilibrium and antioxidant levels that were connected with the reduction of contractility of uterine muscle (Oreščanin-Dušić 2018) logically led us to investigate further on this model. References:
Appiah I, Milovanović S, Radojičić R, Nikolić-Kokić A, Oreščanin-Dušić Z, Slavić M, Trbojević S, Skrbić R, Spasić M, Blagojević D: Hydrogen peroxide affects rat uterine contractile activity and endogenous antioxidative defence. Br. J. Pharmacol. 158, 1932–1941 (2009)
Oreščanin Z, Milovanović SR: Effect of L-arginine on the relaxation caused by sodium nitroprusside on isolated rat renal artery. Acta Physiol. Hung. 93 (4), 271–283 (2006)
Mijušković, A., Oreščanin-Dušić, Z., Nikolić-Kokić, A., Slavić, M., Spasić, M.B., Spasojević, I., Blagojević, D. Comparison of the effects of methanethiol and sodium sulphide on uterine contractile activity (2014) Pharmacological Reports, 66 (3) pp. 373-379.
Mijuškovic, A; Kokić, AN; Dušić, ZO; Slavić, M; Spasic, MB; Blagojević,D. Chloride channels mediate sodium sulphide-induced relaxation in rat uteri. BRITISH JOURNAL OF PHARMACOLOGY, 172 (14):3671-3686; SI 10.1111/bph.13161 JUL 2015.
Zorana Oreščanin-Dušić, Nikola Tatalović, Teodora Vidonja-Uzelac, Jelena Nestorov, Aleksandra Nikolić-Kokić, Ana Mijušković, Mihajlo Spasić, Roman Paškulin, Mara Bresjanac, Duško Blagojević The Effects of Ibogaine on Uterine Smooth Muscle Contractions: Relation to the Activity of Antioxidant Enzymes. Oxidative Medicine and Cellular Longevity Volume 2018, Article ID 5969486, 10 pages, https://doi.org/10.1155/2018/5969486
3 The chemical structures of the active ingredients of ibogaine are needed.
In our experiment, we used pure ibogaine hydrochloride dissolved in water. In this experiment we used ibogaine hydrochloride (PubChem CID: 197059) purity 99.9% (Remøgen, Phytostan Enterprises Inc., Canada) dissolved in deionized water. The reason for the use of ibogaine hydrochloride is its greater solubility in water compared to pure ibogaine.
Ibogaine is a monoterpenoid indole alkaloid, an organic heteropentacyclic compound and an aromatic ether (PubChem CID: 197060). The first sentence in Introduction section is complemented by this information.
In Material and Methods section, subsection Chemicals, we had already provided Compound ID number i.e. identifier from database of chemical molecules and their activities in biological assays, in the sentence: “Ibogaine hydrochloride (PubChem CID: 197059) purity 99.9% (Remøgen, Phytostan Enterprises Inc., Canada) was provided…”
Chemical structures (among other) of ibogaine and ibogaine hydrochloride are presented on respective pages on PubChem: https://pubchem.ncbi.nlm.nih.gov/compound/197060 and https://pubchem.ncbi.nlm.nih.gov/compound/197059.
4 The objective(s) of the work is not clearly mentioned in the abstract and introduction.
We tried to be more clearly, changed some sentences, added some parts about ibogaine action and discussed in more details. We expanded the manuscript and tried to approach this issue.
5 Figure 4 and Figure 5 are somewhat unclear and clearer pictured will be easier to follow the data by readers.
Edited figures are included in revised manuscript.
6 A mechanistic figure and related explanations are required to summarize the data.
Added.
7 What is/are the future work plan with this agent?
Our work plan is to explore do the same mechanism(s) are similar (universal?) in ex vivo models of isolated ileum and blood vessels, as well as ibogaine effects in vivo on different tissues and other antioxidant cellular redox active molecules (thiols). Also, the nature of posttranslational modifications of antioxidant enzymes after ibogaine treatment has to be checked. In addition, the role of calcium and its receptors and signaling pathways in ibogaine activity is also interesting.
In general, ibogaine offers different possibilities, but reports on ibogaine beneficiaries have been ambiguous and opposite. Beneficial effects are evident, but with serious precaution and medical supervision. Our results did not show any serious toxic and harmful ibogaine effects on rats treated with per os doses that did affected its energetic metabolism and redox balance (Vidonja Uzelac 2019a, 2019b, referenced in the text, and Tatalović et al, 2021 „Ibogaine has sex specific effects on the cellular antioxidant systems in rats: the study on females“ submitted for publication). Here, in parallel, we have been trying to explore the mechanisms of its action on ex vivo models that combined ibogaine pharmacological, metabolic and redox properties that contribute and could lead to its both safe and beneficial use.
Round 2
Reviewer 3 Report
Comments were addressed.
Author Response
Dear Reviewer,
Thank You for the suggestion.
Spell checking was performed and some errors were corrected.